# Evidence for pre-climacteric activation of AOX transcription during cold-induced conditioning to ripen in European pear (*Pyrus communis* L.)

**Christopher Hendrickson[1], Seanna Hewitt[1,2], Mark E. Swanson[3], Todd Einhorn[4], Amit Dhingra[1,2]***

**1** Department of Horticulture, Washington State University, Pullman, WA, United States of America,
**2** Molecular Plant Sciences Program, Washington State University, Pullman, WA, United States of America,
**3** School of the Environment, Washington State University, Pullman, WA, United States of America,
**4** Department of Horticulture, Michigan State University, East Lansing, MI, United States of America

* adhingra@wsu.edu

## Abstract

European pears (*Pyrus communis* L.) require a range of cold-temperature exposure to induce ethylene biosynthesis and fruit ripening. Physiological and hormonal responses to cold temperature storage in pear have been well characterized, but the molecular underpinnings of these phenomena remain unclear. An established low-temperature conditioning model was used to induce ripening of 'D'Anjou' and 'Bartlett' pear cultivars and quantify the expression of key genes representing ripening-related metabolic pathways in comparison to non-conditioned fruit. Physiological indicators of pear ripening were recorded, and fruit peel tissue sampled in parallel, during the cold-conditioning and ripening time-course experiment to correlate gene expression to ontogeny. Two complementary approaches, Nonparametric Multi-Dimensional Scaling and efficiency-corrected 2-(ΔΔCt), were used to identify genes exhibiting the most variability in expression. Interestingly, the enhanced alternative oxidase (AOX) transcript abundance at the pre-climacteric stage in 'Bartlett' and 'D'Anjou' at the peak of the conditioning treatments suggests that AOX may play a key and a novel role in the achievement of ripening competency. There were indications that cold-sensing and signaling elements from ABA and auxin pathways modulate the S1-S2 ethylene transition in European pears, and that the S1-S2 ethylene biosynthesis transition is more pronounced in 'Bartlett' as compared to 'D'Anjou' pear. This information has implications in preventing post-harvest losses of this important crop.

## Introduction

The fruit is a specialized organ unique to angiosperms that provides a protective environment for the seeds to develop and mature. In order for the seeds to be disseminated, the fruits undergo a highly-orchestrated set of physiological and biochemical processes that result in

**Data Availability Statement:** All relevant data are within the paper and its Supporting Information files.

**Funding:** The authors thank Blue Star Growers (Cashmere, WA USA) for providing fruit used for this study and to D. Scott Mattinson for assistance in the maintenance of the experimental infrastructure. Work in the Dhingra lab was supported in part by Washington State University Agriculture Center Research Hatch Grant WNP00011 and grant funding from Pear Bureau NW to AD. SLH acknowledges the support received from ARCS Seattle Chapter and National Institutes of Health/National Institute of General Medical Sciences through an institutional training grant award T32-GM008336. The contents of this work are solely the responsibility of the authors and do not necessarily represent the official views of the NIGMS or NIH.

**Competing interests:** The authors have declared that no competing interests exist.

senescence or ripening [1, 2]. The process of ripening is characterized by the breakdown of chlorophyll and accumulation of anthocyanins or carotenoids and xanthophylls; the resulting vivid colors make the fruits visually appealing to potential seed dispersers [3]. The accompanying evolution of aromatic and volatile compounds, conversion of starches to sugars and softening of the mesocarp or cortical tissue make the fruits attractive to consumers [4]. The ripening process is categorized as 'climacteric' when there is a respiratory burst along with a peak in ethylene production [5]. All other modes of ripening that do not demonstrate this characteristic behavior are categorized as 'non-climacteric.' While the latter mode of ripening is represented by various fruits such as citrus, strawberry (*Fragaria × ananassa*), grapes (*Vitis sp.*), etc., the climacteric mode of ripening is exemplified by bananas (*Musa sp.*), tomato (*Solanum lycopersicum* L.), apple (*Malus* x *domestica* Borkh.) and pear, to name a few.

In climacteric fruit, the biochemistry of ethylene biosynthesis is well understood [6, 7]. As a result of enhanced auto-stimulatory production of ethylene during respiratory climacteric, referred to as System 2 ethylene synthesis, the fruit develops a complete profile of desirable sensory qualities for consumption [8–10]. This is accomplished by the activity of ethylene-precursor synthesizing and ethylene-synthesizing enzymes, ACC SYNTHASE (ACS) and ACC OXIDASE (ACO), respectively. In addition to their modulatory effects, these rate-limiting enzymes are themselves under extensive regulation at the transcriptional, post-transcriptional and post-translational levels as demonstrated in banana, tomato, etc. [11–14]. Through a combination of physiological and molecular analyses of climacteric fruit, a comprehensive model of S1 to S2 transition is emerging [5, 15–19] involving numerous phytohormones and molecular signals.

Differential abundance of the ABA catabolic gene transcripts ABSCISIC ACID 8'-HYDROXYLASE 1 and 2 (CYP707A1 and CYP707A2) was shown to correlate with the upregulation of ACS transcripts and specific developmental stage [20]. These genes inhibit expression of NCED-like genes in strawberry and tomato, thereby reducing ABA biosynthesis and promoting cell wall breakdown and ripening [19]. Similar work in peach showed a correlation between endogenous ABA levels with sensitivity to chilling injury and regulation of induction of fruit ripening [17, 21]. Auxin is involved in modulating acute and long-term cold exposure in plants [22, 23]. In climacteric Japanese plum (*Prunus salicina* L.) and melting-flesh peach (*Prunus persica* L.), development and ripening coincides with prolonged cold temperature exposure and changes in auxin metabolic processes, indicating that cold responses in fruit tissues may be influenced in part by intracellular auxin concentrations, though species and cultivars vary in sensitivity [24, 25]. This, in turn, may be controlled by transport, conjugation, biosynthetic, and catabolic mechanisms [23]. For example, the ethylene-signaling repressors EIN3 BINDING FACTOR 1 and 2 were down-regulated in tomato in response to exogenous auxin treatment, thereby propagating the ethylene signal [26]. Similarly, ACS expression was reported to increase in banana fruit upon exogenous auxin application [27].

In addition to phytohormonal regulation, The MADS-BOX TRANSCRIPTION FACTOR Rin (MADS-RIN) protein has long-been considered essential for ripening in climacteric fruits, and RIN binding motifs have been identified in the promoter regions of many genes involved in ethylene biosynthesis and response [28]. Recently, gene-editing based reevaluation of the role of RIN demonstrated that mutation of this gene results in the production of a protein that actively inhibits ripening induction [29]. During ripening, the RIN is known to form a transcriptional regulatory complex that recruits numerous other proteins including specific APETELA 1-like (AP1-like) and SQUAMOSA PROMOTER BINDING PROTEIN-like (SBP-like) in the activation of downstream ETHYLENE RESPONSE FACTOR (ERFs), including those leading to altered ACS or ACO gene expression and protein accumulation [11]. Studies in banana and tomato have further elucidated the components of this transcriptional activation

complex, thereby adding to the understanding of its function, and the functions of additional MADS-box proteins in maturing fruits [30, 31]. Recently, the MADS-RIN protein has been implicated in cold-induced ripening of 'Bartlett' pear [32].

Climacteric respiration primarily consists of the combined activity of cytocyrome c electron transport and electron transport via the alternative oxidase (AOX) pathway, both of which take place in the mitochondria. The alternative respiratory pathway provides a secondary shunt for electrons at times in which respiratory demands are high and the cytochrome pathway is at full capacity. As such, AOX activity prevents overreduction of the ubiquinone pool and serves to help maintain cellular redox homeostasis. Several studies in climacteric fruit have suggested that alternative oxidase activity affects ripening through the propagation of a mitochondria-derived signal [33–35]. This signal is believed to be initiated as a result of imbalance in the cellular redox state when presence of reactive oxygen species is increased, often under conditions of stress and in the presence of ethylene signaling [36]. Furthermore, several studies implicate crosstalk between ethylene, ROS signaling, and AOX activity in maintenance of metabolic homeostasis [37]. In mango, the climacteric stage is facilitated by the up-regulation of cytochrome chain components, and AOX transcript and protein abundance increase after the climacteric peak, reaching a maximum when the fruit is ripe [38]. Moreover, stimulation of AOX by exogenous pyruvate enhanced apple respiration via the alternative respiration pathway at climacteric under cold storage [39]. Similar observations have been recorded in banana, cucumber, and tomato where cold treatment enhanced AOX abundance [40–42]. Collectively, these studies demonstrate that AOX is a product of post-climacteric events and contributes to senescence after the ripening phase.

The transition from the autoinhibitory System 1 to autocatalytic System 2 ethylene production occurs naturally during the developmental course of many fruits such as tomato, apple, peach, and banana. The postharvest cold treatment has been shown to enhance and synchronize production of System 2 ethylene in some pome fruits such as 'Conference' pear and 'Golden Delicious' and 'Granny Smith' apple [43, 44]. Pre- or post-harvest exposure to cold has also been shown to induce ethylene production, and fruit softening in avocado (*Persea americana*) and kiwifruit (*Actinidia deliciosa*) [45, 46]. However, in some cultivars of European pear (*Pyrus communis* L.), a period of exposure to cold temperature after harvest, also called as pre-ripening period, is required for induction of System 2 ethylene production [47, 48]. Such post-harvest cold exposure to ripen fruit has been termed 'conditioning' [49–51]. Previous studies have characterized the conditioning needs for different pear cultivars in terms of storage temperatures, exogenous ethylene exposure and preharvest treatments [49, 50, 52, 53]. In the absence of exogenous ethylene, conditioning can be achieved by storing the fruit at -1˚ to 10˚C for 1–15 days for 'Bartlett' to 60 days for 'D'Anjou' to 90 days for 'Passe Crassane' [51]. At the other end of the spectrum are several Japanese pear (*Pyrus pyrifolia*) cultivars which have no conditioning requirements and are regarded as non-climacteric. In this context, *Pyrus* displays a spectrum of fruit phenotypes in terms of response to cold, induction of S2 ethylene, and the onset of ripening [20]. Interestingly, exogenous application of ethylene can either replace or reduce the need for conditioning and initiate the ripening climacteric, suggesting the existence of cold-induced regulatory processes that act independently of the S1-S2 transition [51].

While general biochemical pathways involved in ripening of model climacteric fruits are well studied, more targeted research is necessary to understand cultivar-specific kinetics and interactions of key ripening-related enzymes in European pears, especially in response to low-temperature conditioning. Previously documented biochemical and genomics data on pear ripening have revealed a complex regulatory crosstalk between numerous phytohormones, secondary messengers, signaling pathways, respiration and chromatin modification [5, 11, 32,

54–56]. Differential regulation of these pathways can generate a spectrum of ripening or post-harvest phenotypes, including delayed or accelerated senescent fruit, and fruit with altered sugar, volatile and nutritional content [15, 57]. Cold-induced physiological responses have been shown to involve various phytohormones such as abscisic acid (ABA), auxin, jasmonic acid, and respiration-related signaling [32].

Several facets of cold-induced ripening in *Pyrus communis* are becoming clearer. Extensive studies in fruit and non-fruit crops have identified the presence of C-REPEAT BINDING FACTOR (CBF)-dependent and independent signaling pathways. A CBF-independent pathway has been shown to modulate the propagation of the cold-signal in various plant tissues through the concerted roles of various phytohormone, messenger, and additional elements [58]. It is expected that phytohormone, respiration and environmental-signaling pathway-targeted analysis of gene expression across European pear cultivars that differ in their cold requirement may reveal insights that integrate and regulate the critical transition from S1 to S2 ethylene production [59, 60].

This study was conducted with a focus on expression changes of key genes involved in ripening-associated biochemical pathways as fruit cultivars representing extreme ends of the chilling requirement spectrum, 'Bartlett' and 'D'Anjou,' underwent physiological conditioning by exposure to predetermined amounts of cold. Nonmetric multidimensional scaling (NMDS) was used to assess relationships between multiple experimental factors of genotype and physiology and the associated expression of key genes. NMDS is a multivariate data reduction technique that identifies axes describing variability among sample units with many measured response variables [61]. The method condenses the many measured variables in a multivariate data set into a reduced number of axes that maximize explained variance. Unlike a number of other methods, however, the method does not require that the measured variables be linear or scaled similarly. This method was used to accommodate the disparity in a large number of data points represented by expression values of individual genes, and a relatively lower number of biological replicates [62]. The NMDS analysis of physiological ripening and expression of target genes revealed that 'Bartlett' and 'D'Anjou' fruits follow two dissimilar vectors in response to cold conditioning, which has implications in preventing post-harvest losses of this important crop.

## Materials and methods

### Physiological conditioning

For this study, fruit was harvested at physiological maturity from two commercial lots in central Washington state. The fruit was obtained within five days of harvest after temporary storage at 1˚C. 'Bartlett' fruit had a mean firmness of 76.2 N, and 13.40˚Brix and 'D'Anjou' fruit had a mean firmness of 53.5 N, and 12.66˚Brix at the time of the initiation of the experiment. Pears of each cultivar were divided into two replicate groups of 1920 fruits each, which were then maintained at 10˚C (Fig 1) for conditioning. Control fruit that did not undergo conditioning were maintained at 20˚C for the entirety of the experiment [49]. After the conditioning period, the fruit was transferred to 180-liter flow-through respiration chambers held at 20˚C for seven days. The flow rate of the chambers was maintained at 5.0 ml/min with compressed air. The fruit was evaluated for firmness at three-time points: at harvest (i.e., 0% conditioning), 100% conditioning, and 100% ripened, which comprised 7 d after completion of conditioning. Tissue samples containing both peel and flesh peeled from the fruit were also collected at the stages mentioned above for subsequent comparative gene expression analysis.

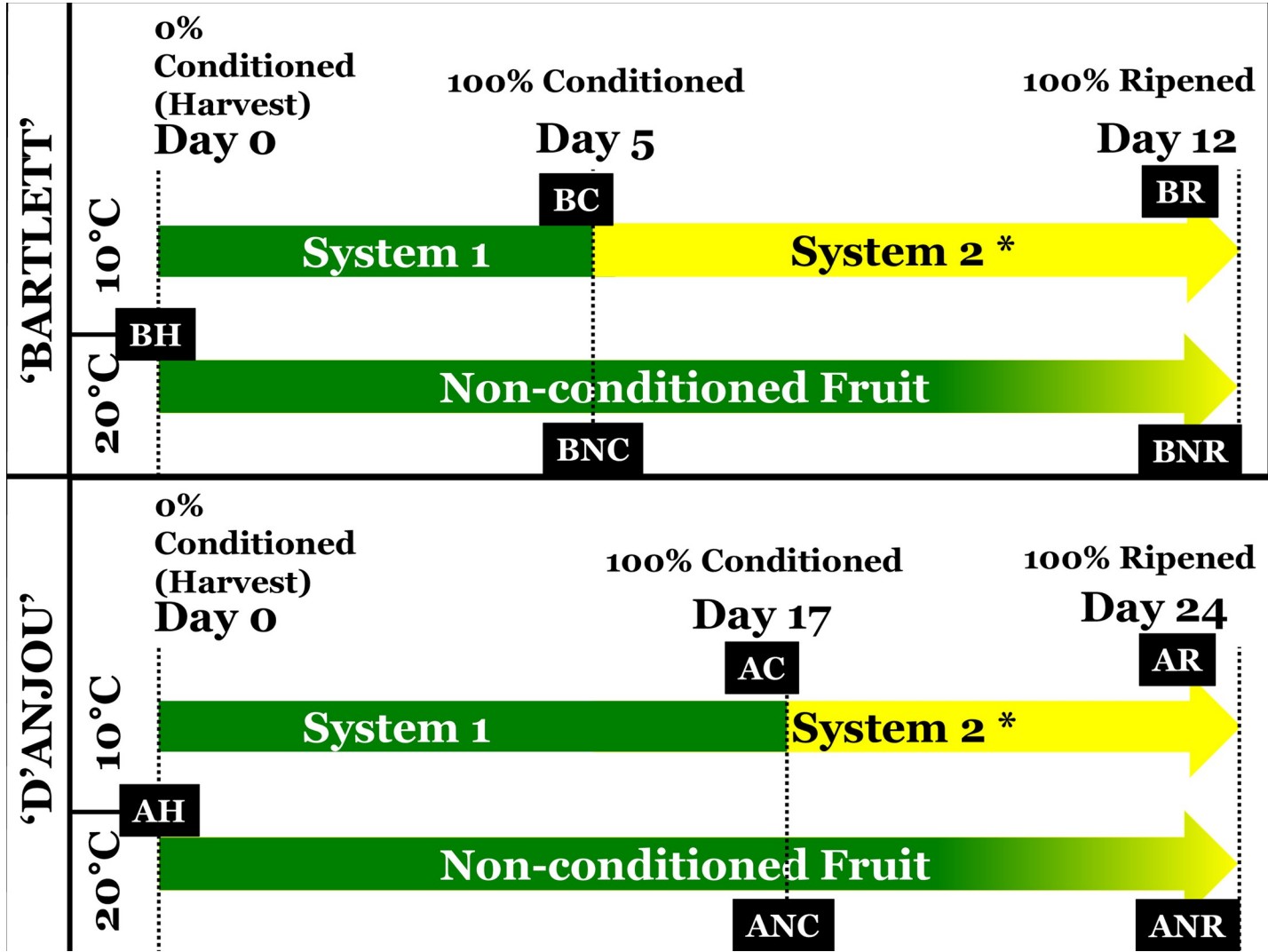

**Fig 1. Treatment and sampling scheme for 'Bartlett' and 'D'Anjou' fruit.** 1,920 fruit of each cultivar were equally distributed. Tissue samples were obtained from Non-Conditioned control fruit maintained at 20°C in parallel to a sampling of fruit that received conditioning treatment. Conditioned fruit was moved to isolated flow-through respiration chambers at 20°C for one week and samples harvested at that time. BH and AH–'Bartlett' and 'D'Anjou' fruit two days after harvest; BC, AC—'Bartlett' and 'D'Anjou fruit at 100% conditioning timepoint; BNC, ANC—'Bartlett' and 'D'Anjou' Non- Conditioned control fruit corresponding to 100% conditioning timepoint for fruit that received conditioning; BR, AR—'Bartlett' and 'D'Anjou' fruit at 100% Ripened stage; BNR, ANR—'Bartlett' and 'D'Anjou' Non-Conditioned control fruit corresponding to 100% ripening timepoint for fruit that received conditioning.

### Fruit firmness

Fruit firmness was measured at each sampling time point, and peel tissue samples were collected from 10 replicate fruit. Firmness was obtained from two equidistant points around the equatorial region of each fruit after removal of the peel with a GS-14 Fruit Texture Analyzer (GÜSS Instruments, South Africa) equipped with an 8.0 mm probe set at 5.0 mm flesh penetration. To determine significant factors impacting changes in fruit firmness, data were assessed using ANOVA, following the statistical approaches described previously [53, 63].

### RNA isolation and cDNA preparation

Peel tissue was obtained from 1 cm wide equatorial region of 3 randomly selected fruit for each treatment and flash-frozen in liquid nitrogen. The tissues were then ground using a SPEX

Freezer/Mill 6870 (Metuchen, NJ USA). Total RNA was extracted from 3 representative time points–harvest, fully conditioned fruit maintained at 10˚C, and fully ripened fruit derived from fruit conditioned at 10˚C following the method described previously [64]. For these time points, corresponding control tissues were also sampled from fruit maintained at 20˚C. For cDNA preparation, RNA samples were treated with DNaseI to eliminate any DNA contamination according to the manufacturer's methods (NEB, Ipswich, MA USA). The RNA concentration was determined for each sample using a Nanodrop ND-8000 (ThermoFisher, MA, USA). The RNA quality was verified using a denaturing gel and BioAnalyzer 2100 (Agilent, CA USA). For each sample, 500 ng of total RNA was used to generate first strand cDNA using random primers provided in the Invitrogen VILO kit (Life Technologies, Carlsbad, CA USA). Product integrity was checked using agarose gel electrophoresis. Concentration for each cDNA preparation was evaluated using a Qubit fluorimeter (Life Technologies–Carlsbad, CA, USA). The samples were diluted to a final concentration of 50 ng/uL. Initial qRT-PCR technical replicate reactions were prepared for each of the 90 selected genes using the iTaq Universal SYBR Green Supermix (BioRad, Hercules, CA). The genes were selected based on a comprehensive literature review to represent phytohormone, secondary messenger and environmental signaling pathways (S3 File summarizes the source of literature used to develop the list of genes involved in S1-S2 ethylene transition and regulation). Reactions were prepared according to the manufacturer's protocols with 100 ng template cDNA. For the amplification phase, samples were denatured at 95˚C for 2:30 min, followed by 50 cycles of 30 s at 95˚C, 30 s at 60˚C annealing temperature and 30 s at 72˚C. For the dissociation phase, samples were denatured for one cycle at 95˚C for 30 s, annealed at 60˚C for 30 s and denatured gradually to 95˚C in increments of 0.5˚C to obtain the dissociation curve.

## Primer design for qRT-PCR

Primers were designed using Primer3 software (http://frodo.wi.mit.edu/) using either the doubled haploid 'Comice' genome [65], or from *Pyrus* ESTs and *Malus × domestica* genome [66] as a template, and were procured from Sigma–Aldrich (St. Louis, MO). The primers were evaluated to ensure single amplicon amplification. Amplicons were gel-extracted, sequenced, then annotated and validated using BLASTX against the NCBI nr database. Primer name, sequences, target gene, and amplicon sequence, are summarized in S3 File.

## Quantitative analysis of targeted gene expression, NMDS analysis

To account for PCR efficiency in the data, Cq values and efficiencies were calculated for each reaction using the LinRegPCR tool [67, 68] (S4 File). Confidence in Cq values resulting from efficiencies below 1.80 or above 2.20 was marked where appropriate by the gene target in further analyses. The Cq values whose efficiency were within these bounds, but exceeded (or equaled) 40.00, were deemed unacceptable and identified in downstream analysis. Similarly, Cq values between (or equal to 35.00–39.99) were marked as '*Low confidence*', where appropriate, by the gene target in further analyses.

Following this, fold-change expression was determined from Cq values of all gene targets (across 4 replicates of all samples) between 'D'Anjou' and 'Bartlett' cultivars using the Pfaffl method [69]. Expression of individual genes was normalized in reference to the geometric mean of *Pyrus communis* β-tubulin and RELATED TO UBIQUITIN1 (RUB1) Cq values, identified as ideal reference genes with NormFinder [70–72] (S6 File). Sequences of these amplicons were determined using Sanger sequencing, then checked for target amplification using BLASTX against the NCBI nr database (S3 File) [73, 74]. This allowed identification of variable expression of individual genes between samples, following methods reviewed previously [75].

**NMDS.**   To determine genes contributing the most to variability in the experiment, Cq values for all remaining gene targets in all biological replicates of all samples were converted into a community matrix (*n* samples by *p* genes) for nonmetric multidimensional scaling (NMDS) using the R package '*Vegan*' [76, 77]. The NMDS process assigns rank-order of each gene expression measurement (Cq in this case) across all samples, then depicts variability in a reduced dimensional space [78]. Graphical or statistical assessment of the grouping of individuals within treatment groups may then be performed to examine dissimilarity in overall gene expression within and among treatment groups. In the present study, these group membership factors included pear cultivar ('Bartlett' and 'D'Anjou'), phenology (harvest, fully conditioned, fully ripened) and conditioning treatment (conditioned, non-conditioned control). To the assess goodness of fit of the final ordination, a stress coefficient was calculated from the data matrix (S5 File).

Within the resulting ordination space (NMDS axis 1 × NMDS axis2), the radial distance of individual gene-associated ordination scores from the origin was calculated and represented as an assessment of contribution of that gene to variability. Pear cultivar membership appeared to be strongly related to NMDS axis 1, while phenology (*harvest*, *conditioned*, *or ripened*) appeared to be strongly related to NMDS axis 2 (S10 File). Sorting radial distance of the plotted points from the vertex produced a list of genes in order of descending contribution to variability. Some additional targets were added to the final list of targets for which additional technical replicates were sought based on *ab initio* and prior unpublished data. From the original set of 90 selected candidates, genes that had the top 25% of joint biplot lengths (radial distance) (S7 File), along with a few additional genes known to be involved in the regulation of ripening were selected. A total of four replicate reactions were performed for 36 gene targets in all biological replicates of all samples. A second two-axis NMDS ordination was performed for 36 targets to visualize variability as a function of each treatment and gene target. A centroid hull plot was generated from expression data among the unique variety-conditioning-phenology treatment combinations in RStudio (raw output in S11 File). Finally, a ray biplot was generated from 12 ABA, auxin, ethylene, cold-signaling and respiration-related genes among this final set to indicate relative contributions of pear cultivar and ripeness to the expression of these highly variable transcripts (raw output in S12 File).

**2- ΔΔCt, Comparative analyses and visualization.**   Cq values were calculated following methods described previously by Pfaffl [79]. In addition to NMDS ordination, fold-change values for individual genes were analyzed to identify highly variable expression at equivalent stages of fruit phenology between conditioned and nonconditioned samples. Gene-by-gene comparisons were conducted after the fold-change data were rendered into a heatmap using a web-accessible tool, Morpheus (raw output as S13 File) [80]. In this study, due to the specific experimental design and limitations of assumptions associated with ANOVA or pairwise t-tests, differences in expression were visualized using a heatmap, and then ranked in a decreasing order before being compared with the results of NMDS ordinations. Genes that showed low-confidence qRT-PCR reaction efficiency or exceeded the parameters described above were marked on the resulting heatmap with an *.

## Results and discussion

In the United States, 97% of the pear orchards exist as low-density plantings with large three-dimensional trees [81–83]. The tree architecture and orchard organization have a significant impact on the physiological quality of the fruit [84, 85]. In order to reduce the extent of variability in fruit quality, fruit used in this study were procured from a commercial warehouse that had been pre-sorted for size. However, it should be noted, that sorting for size does not

necessarily control for variability in the physiological maturity of the fruit, which is affected by canopy position [85]. The conditioning treatment provided to fruit resulted in uniform ripening as evident from changes in fruit firmness, as demonstrated previously [51].

## Fruit firmness

Cold conditioning of the fruit at 10˚C resulted in a reduction of fruit firmness in both 'Bartlett' and 'D'Anjou' cultivars as was demonstrated previously [49, 53, 86]. For both cultivars, fruit softening accelerated once the fruit was transferred to 20˚C. The rate of softening was more rapid for 'Bartlett' than 'D'Anjou' (Fig 2A). Ripening of 'Bartlett' requires 15 d of cold conditioning, while 'D'Anjou' typically requires 60 d of -1˚C to attain ripening competency [87, 88]. The duration of cold conditioning, however, was reduced when conditioning temperatures were increased to 10˚C [49]. 'D'Anjou' pears at advanced physiological maturity stages, achieved through delayed harvest, also ripened with markedly shorter conditioning periods [49]. The rate of softening showed high variability throughout ripening, particularly during the post-conditioning week while the fruit was maintained at 20˚C (Fig 2, S1 File). Conversely, fruit that did not receive cold conditioning, particularly D'Anjou' fruit, failed to soften appreciably (Fig 2B).

## Quantitative analysis of ripening-related genes and NMDS analysis

A comprehensive literature review was used to shortlist 90 key ripening-related genes. The genes represented phytohormone, secondary messenger, and environmental signaling pathways. The expression of these genes was analyzed for both cultivars at different temporal stages during conditioning and ripening. A large number of gene targets in comparison with a limited number of biological replicates, as in most gene expression studies, presents a key challenge to the dimensionality of the experiment [62]. One methodological solution to this challenge is the use of data reduction methods such as ordination, in which the information encoded in many independent variables is distilled into a few dimensions that maximize explained variation [89]. The NMDS procedure assigns rank-order of the measurement associated with each unique treatment combination, then depicts variability in a dimensional space that displays dissimilarity between samples [76, 78, 90]. This approach allows for visualization and pattern recognition among data from numerous samples, including the potential influence of experimental treatment variables. To assess goodness of fit of the final ordination, a stress coefficient is calculated from the data matrix, representing the variability (dissimilarity and dispersion) captured by the set $n$ dimensions (S8 and S9 Files).

The premise of this study, based on non-uniform fruit, warranted a reconsideration of the statistical tests to be applied to the results and their subsequent interpretation. Recent studies have suggested for a need to revisit the interpretation of ANOVA, and pairwise $t$-tests applied to biological data, including fruit texture analysis, gene expression and others [75, 91–93]. To provide physiological context and following protocols applied in prior studies, traditional ANOVA was utilized to identify significant treatment factors in flesh firmness data [63]. Some of these reports focus on the identification of differential expression from microarray data, which features much higher replication than the analytical qRT-PCR data utilized in this study. Indeed, analysis of qRT-PCR data and identification of significant variance in transcript abundance between samples remains challenging due to assumptions inherent to traditional ANOVA and $t$-tests. Specifically, these tests assume equal heterogeneity of variance across biological replicates. Prior reports have demonstrated variable maturation and phenology between fruit depending on location in the canopy, harvest time, and other factors which would introduce large amounts of variance in the fruit, and cDNA preparations derived from

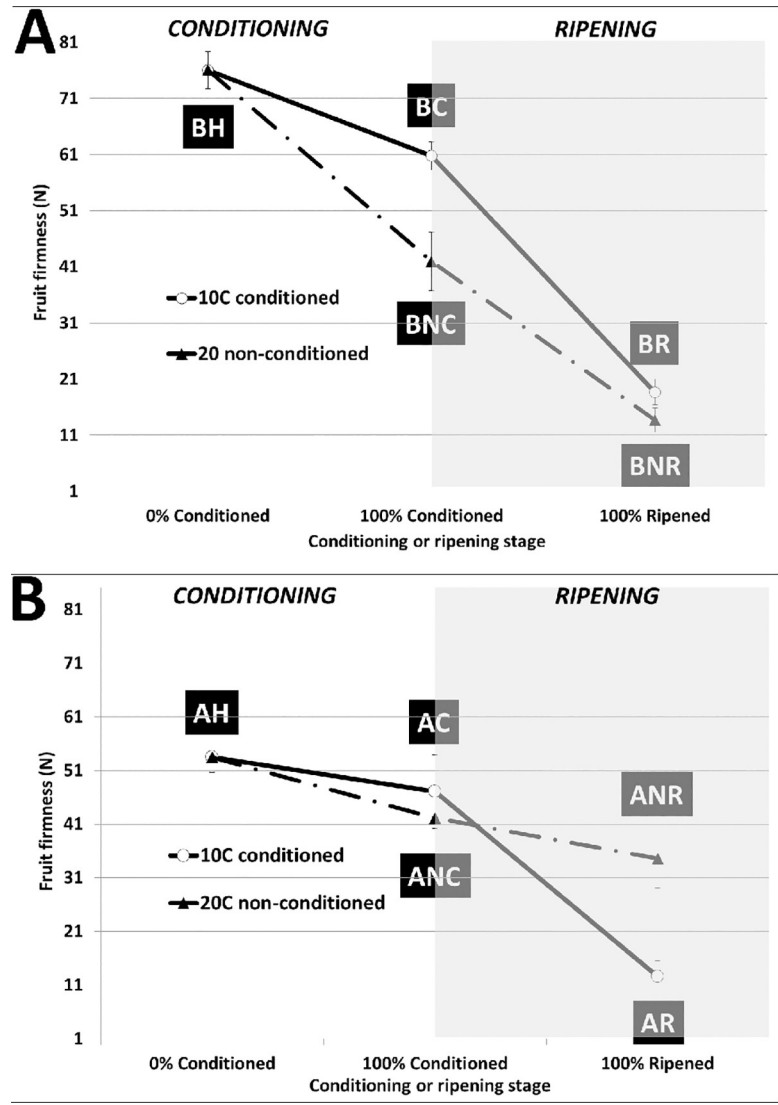

**Fig 2. Mean fruit firmness (N) through conditioning (white background) and ripening (grey background) in A. 'Bartlett', and B. 'D'Anjou' pear.** Fruit was placed into conditioning two days after harvest. Error bars represent standard deviation from the mean of measurements recorded from 10 replicate fruit. Black boxes correspond to fruit treatment sampling stage as follows: BH and AH–'Bartlett' and 'D'Anjou' fruit two days after harvest; BC, AC—'Bartlett' and 'D'Anjou' fruit at 100% conditioning timepoint; 'BNC, ANC—'Bartlett' and 'D'Anjou' Non- Conditioned control fruit corresponding to 100% conditioning timepoint for fruit that received conditioning; BR, AR—'Bartlett' and 'D'Anjou' fruit at 100% Ripened stage; BNR, ANR—'Bartlett' and 'D'Anjou' Non-Conditioned control fruit corresponding to 100% ripening timepoint for fruit that received conditioning.

them [85, 94, 95]. In order to account for the variability, several methods have been proposed including the TREAT-method (*t*-tests relative to a threshold), which frames p-values derived from *t*-tests against an *ab initio*-derived, biologically meaningful point of significance [91]. More recent responses to these challenges propose that inclusion of graphical-estimate approaches may complement, or supplement traditional null-hypothesis based statistical testing, [93]. In this study, nonmetric multidimensional scaling (NMDS) was used for the visualization of broad trends in fruit phenology, as well as the contribution of individual genes in the context of cultivar and treatment.

Nonlinear approaches such as those used in the analysis of qRT-PCR expression data have been used in prior work. Olsvik et al., visualized gene expression with Principal Component Analysis (PCA), also a nonparametric approach, to identify optimal reference genes from a list of targets [96]. This enabled rapid, intuitive selection of those targets exhibiting minimum global variability in the examination of responses in oceanic fish, a heterogeneous environment imparting high experimental dimensionality. Calculation of this global variability is also used in the NormFinder tool described above in the initial analysis of qRT-PCR data. Similarly, a 'progress curve' fitting approach has been proposed in which all fluorescence data points of all reactions are utilized in fitting to the sigmoidal or logistic-growth curve model [97]. NMDS was selected to assess patterns in gene expression with no assumptions of linearity in the data, similar scaling of expression values, or other constraints associated with several other ordination approaches. The reduction of many dimensions associated with a large number of measured genes to a much-reduced number of ordination axes allows the efficient exploration of variability within and among groups.

The initial NMDS ordinations achieved stability after 20 iterations, resulting in stress values ranging from 0.18–0.20 and capturing over 95% of the variability in gene expression data by the pear cultivar and phenology axes [*NMDS axis 1* (graphically inferred to be associated with cultivar) and *NMDS axis 2* (graphically inferred to be associated with phenology)] (S5 File). This indicated that substantial variability among the expression dataset was represented by cultivar and conditioning phenology (Fig 3).

Prior studies have reported a correlation between phenology and the expression of ripening related genes [56, 98, 99]. However, NMDS analysis provides an approach that can capture major axes of variance within a multivariate data set, regardless of the scaling of the variables, and allow for interpretation of the sources of variability. Sorting radial distance of variability in the expression of genes (according to NMDS axes 1 and 2) revealed numerous phytohormone and cold-signaling and additional genes in the approximate top third (S10 File). Further, expression data sorting revealed a tendency to form clusters by cultivar and treatment factors. A rightward-shift is seen in 'Bartlett'-derived expression data in initial and final ordination spaces, relative to 'D'Anjou'-derived expression values. Expectedly, unconditioned controls occupied different regions in the ordination space relative to conditioned fruit of the same cultivar. While unconditioned 'Bartlett' samples remained stationary along the NMDS axis 1 (cultivar), they grouped to the right of (higher axis 1 score) conditioned fruit. Alternatively, unconditioned 'D'Anjou' samples reverted to the left of (lower axis 1 score) conditioned samples. Together, this pattern in the ordination space reveals that there is a pre-existing genotypic variation between 'Bartlett' and 'D'Anjou' fruit regardless of conditioning treatments. This information could be used to devise more efficient, cultivar-specific, conditioning strategies to optimize fruit ripening and quality.

Cq values from ripened fruit samples were generally higher on axis 2 (ripeness) of the ordination space, possibly indicative of the shared physiological responses in the fruit of each cultivar once they acquired ripening capacity post-conditioning. However, 'D'Anjou' fruit exhibit different overall transcriptional responses compared to 'Bartlett' tissues after full conditioning. Of all the transcripts probed in this work, those from 'Bartlett' tissues exhibited a decline in their values on axis 2 of the ordination space initially, followed by a large increase when the fruit begins to ripen during the S1-S2 transition. However, 'D'Anjou' tissues remain comparatively stable for axis 2 in the ordination space, suggesting underlying differences in conditioning response of this cultivar as shown in Fig 3. 'BH' and 'AH'–'Bartlett' and 'D'Anjou' fruit at harvest represents the fruit when the experiments were initiated; BC, AC—'Bartlett' and 'D'Anjou fruit at 100% conditioning timepoint; 'BNC, ANC—'Bartlett' and 'D'Anjou' Non-Conditioned Fruit corresponding to 100% conditioning timepoint (see Fig 1);

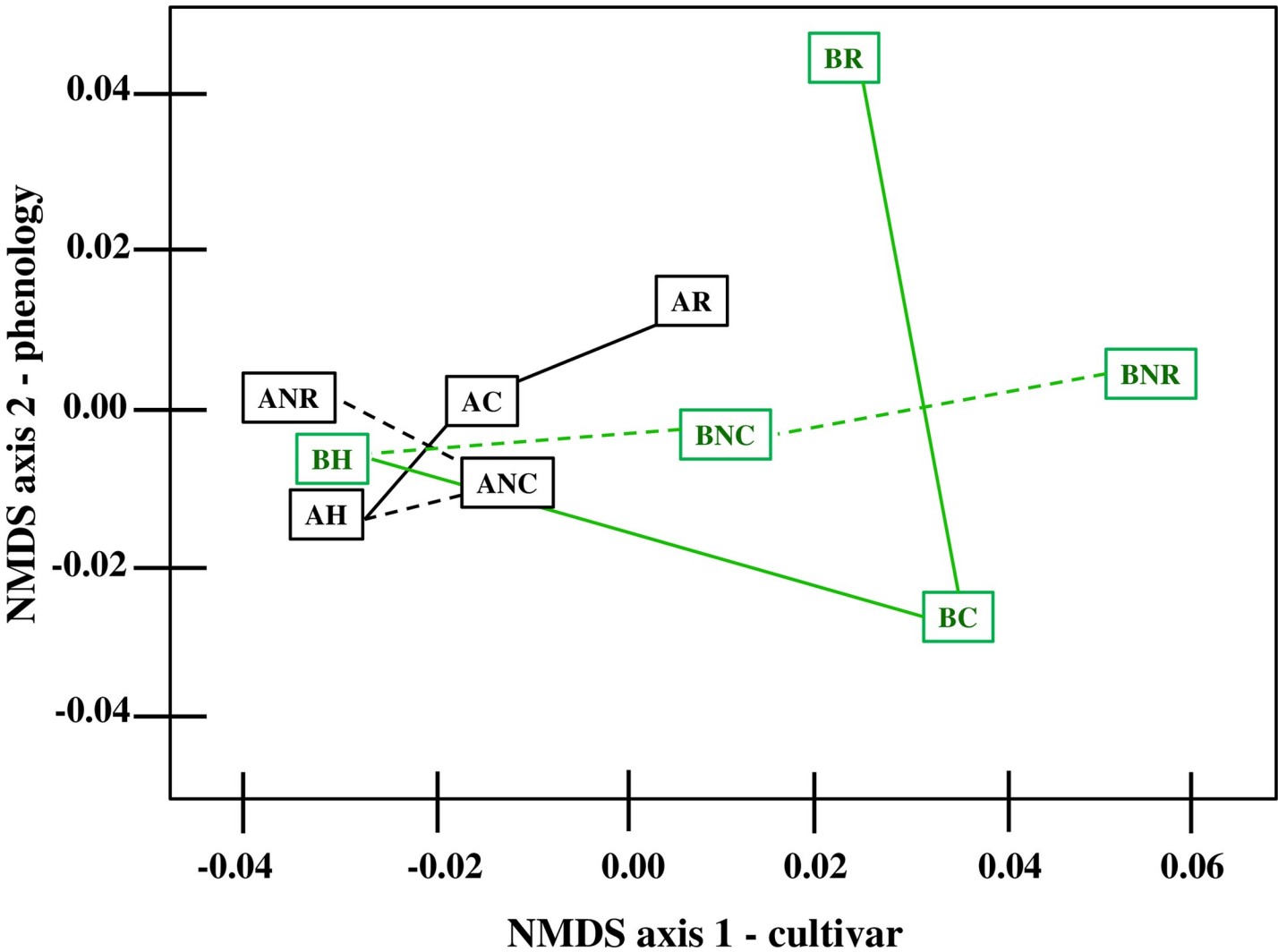

**Fig 3. Geometric treatment group hull centroid plot from final NMDS ordination representing a grouping of expression data according to pear cultivar, ripeness and conditioning treatment in the two-dimensional NMDS ordination space of *x*-axis and *y*-axis (correlating to cultivar and phenology, respectively).** Dashed lines indicate unconditioned control fruit held at 20˚C while solid indicate samples given conditioning treatment. Black lines indicate 'D'Anjou' while green lines indicate 'Bartlett'. BH and AH–'Bartlett' and 'D'Anjou' fruit two days after harvest; BC, AC—'Bartlett' and 'D'Anjou' fruit at 100% conditioning timepoint; 'BNC, ANC—'Bartlett' and 'D'Anjou' Non- Conditioned control fruit corresponding to 100% conditioning timepoint for fruit that received conditioning; BR, AR—'Bartlett' and 'D'Anjou' fruit at 100% Ripened stage; BNR, ANR—'Bartlett' and 'D'Anjou' Non-Conditioned control fruit corresponding to 100% ripening timepoint for fruit that received conditioning. Graph recreated from raw NMDS output in R with Microsoft PowerPoint. Original R output available in S11 File.

BR, AR—'Bartlett' and 'D'Anjou' fruit at 100% Ripened stage; BNR, ANR—'Bartlett' and 'D'Anjou' Non-Conditioned Fruit corresponding to 100% ripening timepoint (see Fig 1). These findings suggest that axis 1 may be discriminating relative 'conditioning need' or 'propensity for S2-ethylene induction and ripening', while axis 2 may discriminate ontogeny of the fruit or relative stage of ethylene production, or ripening.

Data from non-conditioned control samples occupied different regions in the ordination space and followed a different vector relative to conditioned fruit of the same variety. While non-conditioned 'Bartlett' samples (see BH, BNC, BNR, Fig 3) remained unchanged along the NMDS axis 1, they grouped to the right of (higher axis 1 score) conditioned and ripened fruit (see BH, BC, BR, Fig 3). This is consistent with their ripening behavior, where prolonged

storage at 20˚C can soften the fruit, but not necessarily ripen it completely. Alternatively, non-conditioned 'D'Anjou' samples reverted to the left of (lower axis 1 score) conditioned samples (see AH, ANC, ANR, Fig 3). These patterns in the ordination space illustrate inherent cultivar-specific differences between 'Bartlett' and 'D'Anjou' fruit before conditioning treatments. Overall, this plot helps visualize differential transcript abundance and provides a basis for understanding how ripening responses manifest in genetically different pear cultivars subjected to cold conditioning. 'Bartlett' pears transitioned from green to yellow as they ripened, while 'D'Anjou' pears generally retained the green peel color.

Observed changes in expression of the selected genes in the ordination space, which represent independent variables of phenology, cultivar and treatment, align well with the physiological and molecular models of ripening [15, 24, 47, 56, 98, 100]. Recently, Nham et al., identified multiple ABA, auxin, and jasmonic acid-related signaling transcripts as potential putative regulators of cold-induced ripening in European pear, supporting the outcomes of this study [32]. Biplot representation of selected gene vectors and relationship to NMDS ordination axes showed clear grouping according to pear cultivar, correlating calculated radial distance of genes from the vertex of the ordination plot (Fig 4).

Following the second NMDS ordination of expression data from the final set of selected gene targets, distinct associations between expression patterns of genes with NMDS axes 1 (cultivar) and 2 (phenology) were observed. This indicates that the NMDS approach is an additional avenue to visualize multiple independent variables in a statistically relevant space to identify the most important elements that contribute to variability. This may be especially relevant for RNAseq studies where the number of analyzed genes will always be overwhelmingly higher than the number of biological replicates.

The emerging pattern after two rounds of NMDS ordination was that the final third of gene targets focused on in subsequent 2-(ΔΔCt)-parametric analysis comprised putative regulatory control points in the following signaling pathways: cold-perception, abscisic acid signaling, ethylene signaling, auxin signaling, mitochondrial or peroxisomal metabolite transport, and respiration-related genes. The expression behavior of selected genes is discussed in the context of the metabolic pathways they participate in during conditioning and ripening.

**Cold-perception and ABA signaling.** Conditioning temperatures for pear typically range between 0–10˚C; this is well within the range of temperatures at which cold-stress is experienced in any other fruit tissues that do not require prolonged cold-conditioning to acquire ripening competency [16]. The range of conditioning requirements of European pear cultivars indicates variation in cold-stress responses, which is manifest via differential phytohormone and secondary messenger transmission of cold-induced signals. The data were analyzed to explore how the propagation of cold signals varied between 'Bartlett' and 'D'Anjou' tissues. In both Arabidopsis and climacteric fruits, cold acclimation requires activation of the CBF cold response pathway. ICE1 and HOS1 (INDUCER OF CBF EXPRESSION 1 and HIGH EXPRESSION OF OSMOTICALLY RESPONSIVE GENES 1, respectively) comprise initial signaling-elements in cold-perception and initiate the downstream CBF-pathway [21, 101, 102]. Relatively stable expression of the cold signal inducers across pear cultivars and in response to conditioning treatments suggests the presence of an alternative cold-responsive pathway compared to other fruit. Potential candidates could be ABA-signaling or a prolonged cold-responsive RARE COLD INDUCIBLE (RCI)-pathway specific to pear [103]. Differential expression of the cold-signaling transcription factor HOS1 was not observed, though it did appear among the genes contributing the most to total variability in expression among the data. Similarly, analysis of expression from initial technical qRT-PCR replicates did not reveal much variability in ICE1 or CBF-like expression. This highlights the important insights gained

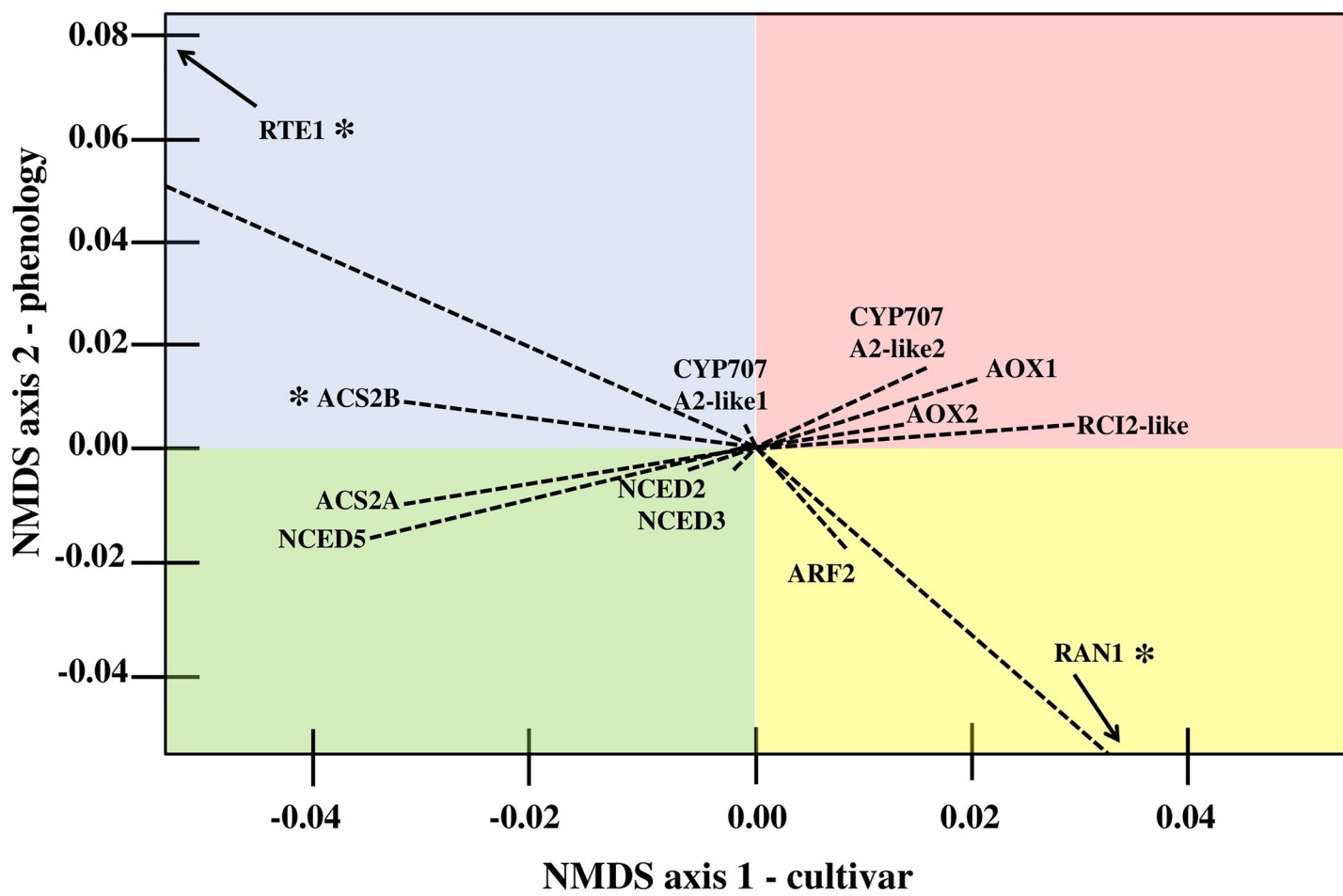

**Fig 4. Ray biplot representation of NMDS axis 1 and axis 2 (correlating to cultivar and phenology on the *x*-axis and *y*-axis, respectively) contributions to expression variability between select gene targets from final NMDS ordination.** Dashed lines represent vector associated with genes in ordination space. Image recreated from raw NMDS output in R. Quadrant shading added to highlight vector separation and is not correlated to, or suggestive of, any physiological state. *-indicates 'low confidence' values, defined as those from which mean Ct equaled or exceeded 35.00, or whose efficiency exceeded 1.80–2.20 in at least one replicate reactions. Raw R output available in S9 File.

into gene expression patterns using a spatial reduction approach, compared to an ANOVA-based comparison of gene expression.

ABA and auxin-pathways have been implicated in cold-induced signaling [104, 105]. Variation of transcript abundance from ABA-biosynthetic and signaling genes was observed in this study. Two pear homologs of ABA-biosynthesis related 9-CIS-EPOXYCAROTENOID DIOOXGENASE (NCED)-like genes were found to associate with the left-side of the NMDS-ordination *x*-axis, corresponding to 'D'Anjou' fruit. Pear CYTOCHROME P450, FAMILY 707, SUBFAMILY A POLYPEPTIDE (CYP707A2)-like expression exhibited slightly divergent orientation in the ordination space. The CYP707A2-vector is extending toward the right of the NMDS axis 1, indicating an enhanced expression in 'Bartlett' throughout the experiment. ABA accumulation may positively influence the transition towards System 2 ethylene biosynthesis and ripening, with steady-state levels significantly impacted by NCED-like, and CYP707A2--like transcript abundance following harvest [20]. A similar impact of ABA accumulation has been reported in peach, where nearly 900 ABA-related differentially expressed genes were correlated with variable cold responsive-phenotypes [21]. Analysis of the data in this study

suggests that ABA biosynthetic activity may vary in pear and depend on the conditioning needs of individual cultivars.

The NMDS ordination suggests that the ABA-biosynthetic NCED-like gene showed a correlation with the increased cold requirement in 'D'Anjou,' while ABA-catabolic CYP707A2--like gene was associated with reduced need for cold in 'Bartlett.' Increased expression of ABA-biosynthetic NCED-like genes and reduced expression of ABA-catabolic genes was shown to correlate with the accumulation of ABA in cold-independent ripening of Asian pear. However, the expression of these genes remains unclear in cold-dependent ripening of Chinese White pear (*Pyrus bretschneideri*) [20], and in 'Braeburn' apple, which also requires cold-exposure to ripen [106]. The results in this study show variable transcript abundance of rate-limiting ABA-biosynthetic genes, providing a critical signature of cultivar-specific cold-response in European pear exposed to established conditioning protocols (Fig 4). Peak abundance of NCED-like and ABA-signaling transcripts showed a correlation to the completion of the conditioning treatment, particularly in 'D'Anjou.' At this phenological stage, 'D'Anjou' fruit had completed the conditioning requirement and had attained full ripening competency, which also marked the induction of S2-associated ACS ethylene biosynthesis [100]. The correlation between ABA-pathway transcript vectors in the ordination space share the trends and position with traditional ethylene-signaling elements, suggesting a collaborative role of these two pathways in regulating cold-induced S2-induction and ripening in European pear.

**Ethylene perception and signaling.** Cold conditioning initiates the S2 transition in European pear, which activates ethylene biosynthesis and downstream signaling processes. In this study, the negative and positive regulators of ethylene-signaling REVERSION-TO-ETHYLE-NE-SENSITIVITY1 (RTE1) and RESISTANT-TO-ANTAGONIST1 (RAN1), respectively, exhibited the largest changes in the biplot vector (Fig 4). The regulatory role of the two genes is evident in the leftward vector associated with RTE1 expression in the NMDS ordination space (Fig 4). Similarly, a RAN1-like transcript exhibited a rightward orientation, supporting the proposed role of this gene in promotion of ethylene signaling, and fruit ripening [107–109]. In Arabidopsis, RAN1 encodes a copper-transporting protein which physically interacts with the ETHYLENE RECEPTOR 1 (ETR1) to deliver the requisite $Cu^{+1}$ ion required for ethylene sensitivity [110]. RTE1 and RAN1 exhibited notable directionality only in unconditioned 'Bartlett,' unconditioned 'D'Anjou,' and fully conditioned 'D'Anjou' samples.

Further, a large increase in fold-change values was observed only in the unconditioned 'D'Anjou' samples. Upregulation of an RTE1-like gene in unconditioned pear would fit with observed physiological responses in pear whose conditioning needs are not met; such fruit would fail to develop ethylene-sensitivity, engage System 2 ethylene biosynthesis or ripen. As a negative regulator of ethylene-signaling, upregulation of an RTE1 like transcript in unconditioned 'D'Anjou' may indicate a repression mechanism in these fruits. There is some evidence for a repressive role for this protein in Arabidopsis where *rte*1 mutants were able to restore ethylene sensitivity in the etr1-2 mutant [111]. In pear, expression of an RTE1-like transcript appears strongly influenced by the stage of conditioning/ripening and cultivar. The magnitude change in RAN1 vector indicates its abundance is strongly impacted by cultivar and phenology, providing the possibility of control of ethylene receptor biogenesis or sensitivity via access to the required copper ion cofactor. The autocatalytic feedback associated with S2 ethylene may be triggered as a result of this altered receptor activity. The pear qRT-PCR product showing homology to RAN1 CDS was sequenced, and its identity confirmed using BLAST (S3 File). A pear RAN homolog was induced at high levels in early ripening, possibly indicating its role in conferring enhanced ethylene sensitivity to the fruit, which is a requirement for S2 ethylene production [112]. Characterization of RAN and RTE1-like homologs in pear may help

determine if these genes indeed exhibit regulatory control over the ripening competency, ethylene signaling or biosynthesis, or cold-signaling.

Pear ACS2-like transcripts exhibited vectors near the vertex of the ripeness axis in the biplot but were located to the left, supporting the role of ACS2 in S1 to S2 transition and ethylene biosynthesis during conditioning [99]. This narrow transitionary period also appears to coincide with increased RARE COLD-INDUCIBLE (RCI)-like expression. The role of RCI-like genes is poorly understood outside of a few model systems. In Arabidopsis, RCI-proteins attenuated ethylene biosynthesis and cold-acclimation by destabilizing ACS proteins. The 14-3-3 protein RCI1 destabilizes all three ACS types in Arabidopsis [113] via a yet to be characterized CBF- and ABA-independent cold signaling pathway. This would agree with the general functional role of 14-3-3 proteins in modulating target protein activity through physical interaction [114]. Similarly, expression of RCI-like transcripts may be regulated through a CBF- and ABA-independent pathway of cold-responsive signal transduction [115]. An RCI2-like transcript oriented toward the right of the NMDS axis 1, in alignment with the recent reports of its role in the promotion of ripening in tomato [103].

**Auxin perception and signaling.** Changes in free auxin concentrations have been positively correlated to ripening induction in many other climacteric fruits [5, 15, 26, 59, 116]. In the climacteric Japanese plum, the seasonal harvest time of fruit was correlated to TIR1-like auxin receptor haplotype [117]. However, expression of a TRANSPORT INHIBITOR RESPONSE 1 (TIR1)-like auxin receptor did not vary significantly in this study. While knowledge from other climacteric fruits suggests cold-adaptive capacity, auxin-sensitivity may be related to specific pear cultivars.

Cold-responses in plant tissues may be attenuated from intracellular auxin concentrations, which in turn may be controlled by transport, conjugation, biosynthetic, and catabolic mechanisms [22]. Recent work in Japanese plum also correlates cold-adaptive capacity to auxin-sensitivity, System 2 ethylene biosynthesis and fruit ripening through a mechanism not yet characterized [117]. The abundance of some, but not all, auxin-signaling related transcripts displayed variation in this study. The vector for a transcript bearing homology to an AUXIN-RESPONSE FACTOR 5 (ARF5) oriented toward the right of the cultivar axis (toward 'Bartlett'), but towards reduced ripeness. This agrees with prior studies where a correlation between ARF-like expression to the regulation of abscisic acid and ethylene-signaling has been demonstrated [12, 118–120]. In Arabidopsis, studies have linked induction of the ARF2 transcriptional activator to ABA, demonstrating the critical relationship between these two phytohormones in mediating the response to environmental cues. In tomato, an ARF2-like homolog functions at the intersection between activities of other phytohormones impacting ethylene, abscisic acid, cytokinins, and salicylic acid signaling [121].

**Transcription and respiration activators.** Climacteric fruit is characterized by a concomitant increase in respiration and ethylene production [48], processes which are expected to be coordinated by various transcriptional activators. Two MADS-box like transcripts were found to be variably expressed between 'Bartlett' and 'D'Anjou' fruit that received conditioning treatment and were held at 20°C. Transcripts bearing homology to the *Malus × domestica* MADS-RIN like transcription factor 8 and 9-like exhibited significant differential expression in response to conditioning and phenology, but not to the cultivar. This suggests that there may be a shared mechanism between 'Bartlett' and 'D'Anjou' during the acquisition of ripening competency that may be mediated by MADS-box transcriptional regulators. The intricate roles of these important regulatory players have been detailed extensively in tomato, and research has described the complex network of interactions and subsequent regulatory influence of the MADS-RIN protein on fruit ripening induction [28, 122]. MADS-RIN/AP1-like genes play a significant role in tomato ripening and are thought to recruit the redundant

FRUITFUL1 and 2 (FUL1 and 2) MADS-box proteins to regulate fruit ripening under ethylene-dependent and independent pathways [123, 124]. Conditioned (and not ripened) 'Bartlett' and 'D'Anjou' samples exhibited far more variability in expression of both transcriptional activators, suggesting that temporal variation in MADS-box gene expression results in differential regulation of the S1-S2 transition among pear cultivars. It seems likely that the expression of this transcription factor may serve as an indicator of pear ripening competency through conditioning treatments, lending support to the growing understanding of the MADS-box/AP1/SBP-transcriptional regulatory complex that acts during the respiratory climacteric.

Interestingly, accumulation of ALTERNATIVE OXIDASE (AOX) 1-like transcripts varied during phenology of 'Bartlett' and 'D'Anjou' samples. Transcript abundance peaked in the "100% Conditioned" stage of both cultivars, which is a pre-climacteric stage, as the fruit is yet to transition to S2 stage [35]. This is intriguing since the expression of AOX has been shown to coincide with the climacteric peak, a characteristic of climacteric fruit [39, 125]. However, AOX-1 expression may have peaked between sampling time points, particularly in initial responses to conditioning environments. These results were not observed for AOX2-like transcripts, differing from trends previously reported in mango and tomato fruit [34, 126, 127].

AOX1 and AOX2-like expression has been reported in many fruit systems, with AOX isoforms displaying responses to a broad range of stresses, including cold-stress. Knock-down *AOX* in tomato delayed ripening, indicating a regulatory role of AOX [128]. Notably, AOX overexpression tomato lines were shown to be far less responsive to the ethylene signaling inhibitor 1-methylcyclopropene (1-MCP), while knock-out lines were highly responsive. Thus, in European pear, respiratory partitioning into the alternative pathway may impact S2 ethylene biosynthesis, the climacteric respiration peak, and consequent ripening-related trait development, independent of prior ethylene sensitivity [34, 129]. A mechanism for the observed variation in AOX transcripts between the tested pear cultivars and other model climacteric systems is unclear, though such variation in AOX expression and activity has been reported in many plants for some time [130, 131].

## Comparative gene expression analyses using LinReg PCR-corrected 2-(ΔΔCt)

The LinRegPCR workflow was applied to the 2-(ΔΔCt) expression data for 27 genes. A heatmap of the relative expression was produced using Morpheus [80], which illustrates variable patterns of selected genes between 'Bartlett' and 'D'Anjou' (Fig 5). Nearly half of the genes included for final analysis exhibited down-regulation throughout fruit phenology in 'Bartlett' and 'D'Anjou' samples relative to harvest samples. The second half of genes exhibited upregulation, exhibiting cultivar or conditioning treatment-dependent deviations.

Variable transcript abundance at the fully conditioned stage was evident for NCED-, ABI3- and ARF-like 'D'Anjou' transcripts (columns 2 and 3 of Fig 5). Similar results were apparent for 'Bartlett' (columns 7 and 8), with variability in response to conditioning evident for FUL1, ACS2, RAN1, ARF5, and CYP707A2-like transcripts. 'Bartlett' tissues exhibited generally higher AOX1 and AOX2 transcript abundance, suggesting that AOX induction in 'D'Anjou' may be muted. Enhanced AOX transcript abundance and activity has been associated with accelerated ripening in other climacteric systems in which S2 ethylene-production is impaired [35, 125]. These results highlight shared and unique transcriptional responses in these two cultivars that have a very different conditioning requirement for the transition to S2.

In this analysis approach, increased AOX-like transcript abundance coincided with that of ACS, ARF, and ABA-related genes for 'Bartlett' and 'D'Anjou' samples, suggesting that these pathways comprise at least part of a coordinated transcriptional cascade that results in S2

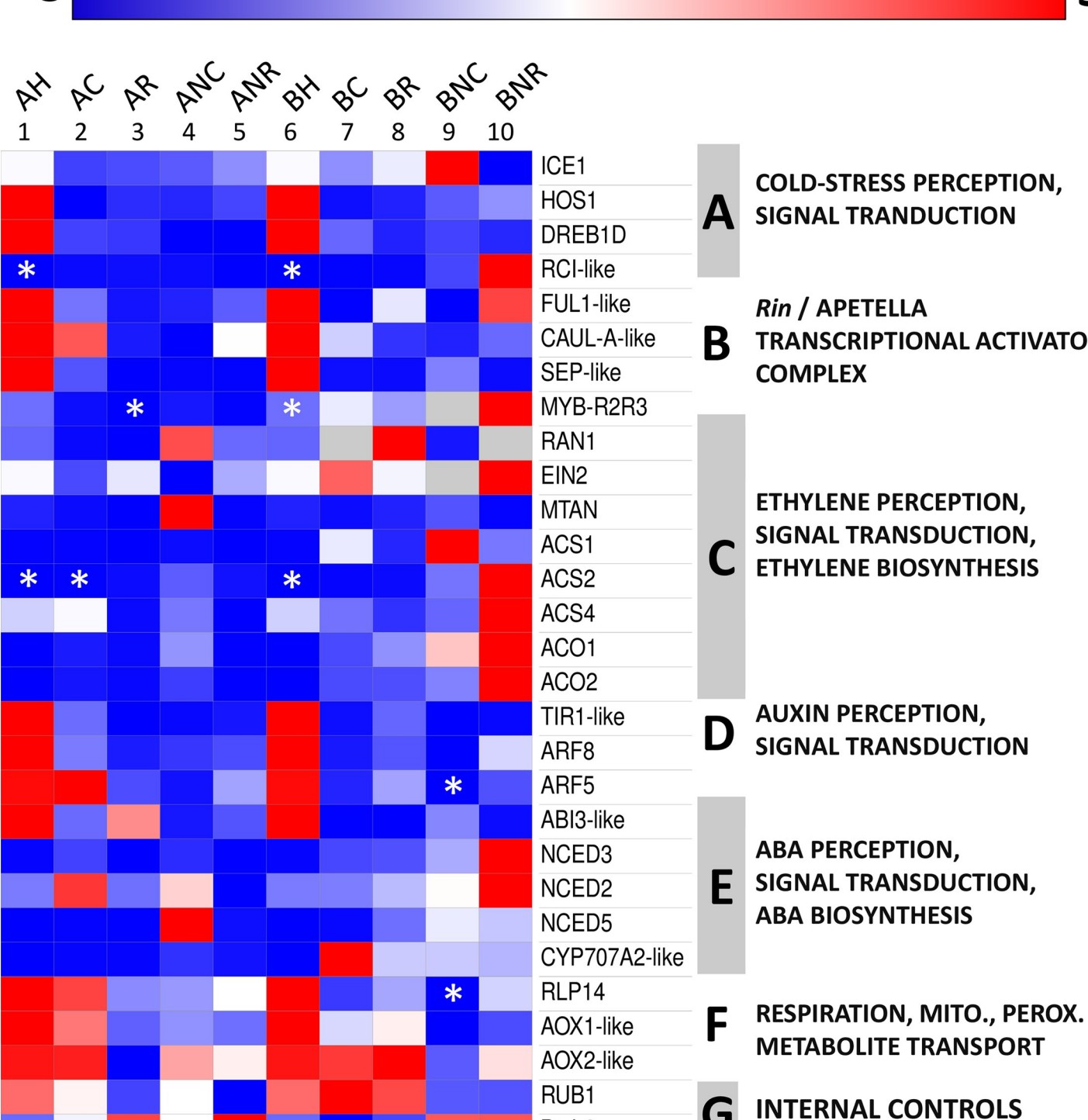

**Fig 5. Fold-change gene expression values (from -3 to +3) of topmost variably expressed genes from the qRT-PCR analysis in D'Anjou (columns 1–5), and Bartlett (columns 6–10), following sorting from final NMDS ordination.** Grey cells indicate that a gene was not detected in the sample. *-indicates 'low confidence' values, defined as those from which mean Cq equaled or exceeded 35.00, or whose efficiency exceeded 1.80–2.20 in at least one replicate reactions. Gene annotations on right side of the heatmap, labeled as letters A-G. A- cold signaling, B- transcriptional regulators, C- ethylene signaling, D- auxin signaling, E- abscisic acid signaling, F- peroxisomal or mitochondrial metabolite transport and respiration-related, G- internal controls. BH and AH–'Bartlett' and 'D'Anjou' fruit two days after harvest; BC, AC—'Bartlett' and 'D'Anjou fruit at 100% conditioning timepoint; 'BNC, ANC—'Bartlett' and 'D'Anjou' Non- Conditioned control fruit corresponding to 100%

conditioning timepoint for fruit that received conditioning; BR, AR—'Bartlett' and 'D'Anjou' fruit at 100% Ripened stage; BNR, ANR—'Bartlett' and 'D'Anjou' Non-Conditioned control fruit corresponding to 100% ripening timepoint for fruit that received conditioning. Heatmap generated with Morpheus with additional sample and pathways annotated using Microsoft Powerpoint. Raw Morpheus output is available in S13 File.

ethylene biosynthesis and acquisition of ripening competency (Fig 5). In this role, induction of the AOX pathway in pear may provide an additional hub of regulatory control beyond the MADS-RIN complex, which integrates signals from cold-signaling pathways, while relieving limited energy production and metabolic flux from the mitochondria. Some initial work in climacteric systems suggests this control point could affect increased ethylene biosynthesis through a retrograde signal derived from respiratory activity, metabolic flux or energy limitation [131]. The cold and phytohormone-responsive transcriptional activator MYB29 was shown in Arabidopsis to be a putative regulator of AOX activity via such a retrograde signaling mechanism, integrating numerous hormonal and signaling pathways with respiration [132]. MYB29-like and other R2R3-MYB genes have only recently been the subject of broad comparative analyses in other climacteric crops [133] and can be found in *P. communis* and *P. bretschneideri* genomes. Such genes were also found to be a target of miRNAs influencing post-cold storage physiological responses in Litchi [134].

Adding to this, sulfur signaling in plants may closely impact respiratory activity, oxidative signaling, ethylene signaling and may interact with nitric oxide pathways [135–138]. Among the most variably abundant transcripts in this work was the plastid-derived Rhodanese-like domain-containing protein 14, a sulfurtransferase localized to the thylakoid. The full complement of the functional relevance of this protein in plants is unclear, though, after full conditioning treatment, more of this transcript was found relative to 'Bartlett' tissue, which generally loses green pigmentation more rapidly upon ripening onset than 'D'Anjou'. Overall, these data, along with results of other recent studies, presents the possibility that in European pear and other climacteric fruit, variation in respiratory and phytohormone-pathway activity is not just a consequence of environmental factors, but also mediates physiological responses to them. Generally, the results from LinReg PCR-corrected 2-(ΔΔCt) correspond to what was observed with the NMDS approach.

Utilizing a targeted gene approach, this study allowed for focused analysis of genes documented to play important roles in cold-induced conditioning and subsequent ripening in pear. It does not, however, capture transcript abundance of the breadth of genes including those regulating S2-ethylene and ripening related chromatin modifications, epigenetic regulation or small RNAs, all of which have been reported to impact these processes in model fruit systems.

## Conclusions

This study adds cultivar specific information regarding the response of European pear to cold conditioning and lends insight into the genetic changes that occur as fruit transitions to S2 ethylene production. Results of this work indicate that cold-sensing and signaling elements from ABA and auxin pathways modulate S1-S2 ethylene transition in European pears, and suggest that, while 'D'Anjou' pear is able to mitigate and cope with the effects of cold exposure, 'Bartlett' is comparatively less-equipped, resulting in a more pronounced S1-S2 ethylene biosynthesis transition. Interestingly, enhanced alternative oxidase transcript abundance in 'Bartlett' and 'D'Anjou' tissues at the peak of the conditioning treatments suggests that AOX plays a novel role in the achievement of ripening competency in European pear.

## Supporting information

**S1 File. 'Bartlett' and 'D'Anjou' pear flesh firmness raw data and ANOVA analysis.**
(XLSX)

**S2 File. Quantitative RT-PCR reaction conditions, thermal profile.**
(DOCX)

**S3 File. Quantitative RT-PCR primer names, primer sequences, (Sanger) sequenced amplicons used and reference providing ab initio annotation to candidate regulatory transcript (s).**
(XLSX)

**S4 File. LinRegPCR input as raw fluorescence readings from the qRT-PCR instrument and PCR cycle, with resulting efficiency and Cq output with regression statistics (Ramakers et al., 2003).** LinRegPCR run in plate-wide mean calculation mode. Data separated by tabs as input and output for each plate run in the qRT-PCR analysis.
(XLSX)

**S5 File. Initial NMDS ordination of fold-change values from initial qRT-PCR reaction replicates from all gene targets.**
(PPTX)

**S6 File. NormFinder candidate reference gene input and output.**
(XLS)

**S7 File. Community master data matrix for initial and final NMDS ordination.**
(XLSX)

**S8 File. Stress plots from initial and final NMDS ordination plot.** Stress plots of the initial (circles) and second (triangles) NMDS ordination procedures. Both instances produced a final stress coefficient of nearly 0.20 after 20 iterations.
(DOCX)

**S9 File. Raw R code, NMDS modeling.**
(TXT)

**S10 File. The radial distance of 90 gene targets from initial NMDS ordination plot vertex, representing total variability as a function of ripeness and pear cultivar.** The radial distance of final 36 gene targets from final NMDS ordination plot vertex, representing total variability as a function of pear cultivar and ripeness (NMDS axes 1 and 2, respectively). For NMDS-2, table of the radial distance of final 36 gene targets from initial NMDS ordination plot vertex, representing total variability as a function of pear cultivar and ripeness (NMDS axes 1 and 2, respectively).
(DOCX)

**S11 File. Raw R output, centroid hull plot from 36 genes following initial NMDS ordination plot representing total Cq variability by treatment group as a function of pear cultivar and ripeness (NMDS axes 1 and 2, respectively).**
(PPTX)

**S12 File. Raw R output, final 12 gene-set vector plot following final NMDS ordination plot from the vertex, representing total Cq variability as a function of pear cultivar and ripeness (NMDS axes 1 and 2, respectively).**
(PPTX)

**S13 File. Raw Morpheus heatmap tool output in PDF format.**
(PDF)

## Acknowledgments

Disclaimer: The contents of this work are solely the responsibility of the authors and do not necessarily represent the official views of the NIGMS or NIH.

The authors thank Blue Bird Growers (Peshastin, WA USA) and Blue Star Growers (Cashmere, WA USA) for providing fruit used for this study, and to D. Scott Mattinson for assistance in the maintenance of the experimental infrastructure. Work in the Dhingra lab was supported in part by Washington State University Agriculture Center Research Hatch Grant WNP00011 and grant funding from Pear Bureau NW to AD. SLH acknowledges the support received from ARCS Seattle Chapter and National Institutes of Health/National Institute of General Medical Sciences through an institutional training grant award T32-GM008336.

## Author Contributions

**Conceptualization:** Amit Dhingra.

**Data curation:** Mark E. Swanson.

**Formal analysis:** Christopher Hendrickson, Seanna Hewitt, Mark E. Swanson, Todd Einhorn, Amit Dhingra.

**Funding acquisition:** Amit Dhingra.

**Investigation:** Christopher Hendrickson, Todd Einhorn, Amit Dhingra.

**Methodology:** Christopher Hendrickson, Mark E. Swanson, Amit Dhingra.

**Project administration:** Amit Dhingra.

**Resources:** Todd Einhorn, Amit Dhingra.

**Software:** Mark E. Swanson.

**Supervision:** Todd Einhorn, Amit Dhingra.

**Validation:** Christopher Hendrickson, Amit Dhingra.

**Visualization:** Mark E. Swanson.

**Writing – original draft:** Christopher Hendrickson, Seanna Hewitt, Amit Dhingra.

**Writing – review & editing:** Christopher Hendrickson, Seanna Hewitt, Mark E. Swanson, Todd Einhorn, Amit Dhingra.

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
