## [Decision Letter · Decision Letter 0]

31 Oct 2019

PONE-D-19-25331

Evidence for pre-climacteric activation of AOX transcription during cold-induced conditioning to ripen in European pear (Pyrus communis L.)

PLOS ONE

Dear Professor Dhingra,

Thank you for submitting your manuscript to PLOS ONE. After careful consideration, we feel that it has merit but does not fully meet PLOS ONE’s publication criteria as it currently stands. Therefore, we invite you to submit a revised version of the manuscript that addresses the points raised during the review process.

We would appreciate receiving your revised manuscript by Dec 15 2019 11:59PM. To enhance the reproducibility of your results, we recommend that if applicable you deposit your laboratory protocols in protocols.io, where a protocol can be assigned its own identifier (DOI) such that it can be cited independently in the future. For instructions see: http://journals.plos.org/plosone/s/submission-guidelines#loc-laboratory-protocols

We look forward to receiving your revised manuscript.

Kind regards,

Muthamilarasan Mehanathan, Ph.D.

Academic Editor

PLOS ONE

Journal Requirements:

"The authors thank Blue Star Growers (Cashmere, WA USA) for providing fruit used for this study and to D. Scott Mattinson for assistance in the maintenance of the experimental infrastructure.  Work in the Dhingra lab was supported in part by Washington State University Agriculture Center Research Hatch Grant WNP00011 and grant funding from Pear Bureau NW to AD. SLH acknowledges the support received from ARCS Seattle Chapter and National Institutes of Health/National Institute of General Medical Sciences through an institutional training grant award T32-GM008336. The contents of this work are solely the responsibility of the authors and do not necessarily represent the official views of the NIGMS or NIH.

Reviewers' comments:

Reviewer's Responses to Questions

**Comments to the Author**

1. Is the manuscript technically sound, and do the data support the conclusions?

Reviewer #1: Yes

Reviewer #2: Yes

2. Has the statistical analysis been performed appropriately and rigorously? 

Reviewer #1: Yes

Reviewer #2: Yes

3. Have the authors made all data underlying the findings in their manuscript fully available?

Reviewer #1: Yes

Reviewer #2: Yes

4. Is the manuscript presented in an intelligible fashion and written in standard English?

Reviewer #1: Yes

Reviewer #2: Yes

5. Review Comments to the Author

Reviewer #1: This is an interesting study, particularly the analysis of the gene expression data.

Some clarification is recommended for the following points:

Line 171: Was the non-conditioning control kept at 20C in the same way as the conditioning treatment?

Line 174: Why was the peel tissue used and not the flesh?

Line 196: What primers were used for cDNA synthesis?

Line 197: Why was the cDNA nanodropped? What information were you expecting to obtain? dNTPs can often interfere with the reading of cDNA.

Line 223: "or above 2.20".

Line 229: How many technical replicates were run for qRT-PCR?

Line 272: What condition was defined as the calibrator (to normalize the expression data)?

Comments about the intro: Since the manuscript is titled: "Evidence for pre-climacteric activation of AOX transcripcion during cold-induced...", I would recommend adding additional background about the role of this gene in the introduction. What it is currently present may not enough.

Reviewer #2: The manuscript entitled “Evidence for pre-climacteric activation of AOX transcription during cold-induced conditioning to ripen in European pear (Pyrus communis L.)” contributes significantly towards understanding of the species- specific fruit ripening process of European pear and has direct implications in preventing post-harvest losses in P. communis. The work is well-performed with defined objectives, appropriate methodology, novel results with statistical significance. The manuscript is well-written with organized figures and tables, and it merits publication with some minor corrections:

1. The sample collected for gene expression studies is unclear. Why the expression has to be studied in the peel tissue and not the other part(s)?

2. In line 172, it is mentioned as ‘The fruit was evaluated for physiological parameters’; however, it is unclear what are the parameters studied? The description is available only for ‘fruit firmness’.

3. It is unclear how many replicates were maintained for qPCR analysis. Also, the statistical treatment corresponding to it has to be described.

4. Although the manuscript is well written, the list of references and the in-text citations are not proper in many places. All the references in the reference list should be as per the journal format. In many positions cited references has not been included in the reference list. Those should be incorporated.

5. Line 109: Place 43 and 44 within the same bracket.

6. In the materials and methods, in many places, reference number has not been mentioned for the in text citations. Line 170: Replace ‘Sugar and Einhorn, 2011’ by reference number 47.

7. In Line 190: the reference ‘Gasic et al., 2004’ is missing in the list of references. In text citation should be numbered.

8. Line: 192: DNAse will be ‘DNase’

9. Line: 222: Both the references: ‘Ramakers et al., 2003’ and ‘Ruijter et al., 2009’ are not present in the list of references. Their corresponding in text citations should be numbered.

10. Line: 231: Pfaffl, 2001: Reference should be corrected and corresponding reference number to be mentioned in the text.

11. Line 233: ‘Andersen et al., 2004; Imai et al., 2014; Vandesompele et al., 2002’ all of these references are not present in the list of references.

12. Line 236: ‘Altschul et al., 1990; Gish and States, 1993’ these references are also missing in the list of references.

13. Line 238: mention the corresponding reference number (#77) for ‘Rieu and Powers, (2009)’.

14. Line: 272: Use symbols for ‘2-delta-delta Ct’

15. Line 301, 303: Mention the reference number (#47) instead of ‘Sugar and Einhorn, 2011’

16. Line 317: ‘Wayland, 2003’ reference missing in the list of references.

17. Line 319: Replace ‘Kruskal, 1964’ by reference # 64, Replace ‘Krzywinski and Altman, 2014’ by reference # 66. The reference ‘Young, 1970’ is missing in the list of references.

18. Line 350: ‘Olsvik, Søfteland’ will be ‘Olsvik et al.’

19. Line 423: ‘Nham, Macnish’ will be ‘Nham et al.’

20. Line 496: Mention the reference # 87 replacing ‘El-Sharkawy, 2004’

21. Line 509: All of the references ‘Chang et al., 2014; Ma et al. 2012; Qiu et al., 2012’ are missing in the list of references.

22. Line 537: Mention the reference # 87 replacing ‘El-Sharkawy, 2004’

23. Line 547: Replace ‘Sivankalyani et al., 2014’ by reference # 90.

24. Line: 565: Replace ‘Liu et al., 2013’ by reference # 12

25. Line 566: Replace ‘Tacken et al. 2012a’ by reference # 58. References ‘Robles et al., 2012; Schaffer et al., 2013;’ are not present in the list of references.

26. Line 618: Mention reference # 68 within ()

27. For the list of references, the formatting should be corrected wherever they are not as per the journal format

28. Line: 727: Rewrite the reference as per journal format

29. Line 835: Rewrite the reference as per journal format

30. Line 880: Rewrite the reference as per journal format

31. Line 916: Rewrite the reference as per journal format

32. Line 925: Correction needed

33. Line 929: Rewrite the reference as per journal format

34. Line 966: Rewrite the reference

35. Line 1012: Rewrite the reference

36. A minor polishing in terms of grammar is required.

6. PLOS authors have the option to publish the peer review history of their article (what does this mean?). If published, this will include your full peer review and any attached files.

Reviewer #1: No

Reviewer #2: No

---

## [Author Response · Author response to Decision Letter 0]

7 Nov 2019

Dear Editor, 

The authors would like to thank both the reviewers for their critical, and very helpful comments. We have incorporated the minor revisions, which has substantially improved the manuscript. All the edits are visible in the manuscript with track changes. It is our hope that the manuscript will be acceptable for publication in its revised form. As advised, we have provided a point-by-point response to the reviewers’ comments. The author’s responses to reviewers’ comments are as follows.

Reviewer #1: 

Reviewer’s comment: This is an interesting study, particularly the analysis of the gene expression data. Some clarification is recommended for the following points:

Response: Thank you. Point by point response to the reviewer’s comments follows.

Reviewer’s comment: Line 171: Was the non-conditioning control kept at 20C in the same way as the conditioning treatment?

Response: The controls fruit were kept at 200C throughout the entirety of the experiment. The experimental design is presented in Figure 1. An additional sentence was added to clarify the experimental design. 

Reviewer’s comment: Line 174: Why was the peel tissue used and not the flesh?

Response: This sentence has been corrected in the text. Since the tissue was peeled, it contained both peel and flesh tissues. Similar tissue has been used for RNAseq work in pears in previous publications (Nham et al. 2015; Nham et al. 2017).

Reviewer’s comment: Line 196: What primers were used for cDNA synthesis?

Response: Random primers that were part of the Invitrogen VILO kit were used for cDNA synthesis. This information has been added to the text. 

Reviewer’s comment: Line 197: Why was the cDNA nanodropped? What information were you expecting to obtain? dNTPs can often interfere with the reading of cDNA.

Response: Thanks for pointing this out. That was an oversight. Both agarose gel electrophoresis and Qubit fluorometer were used for evaluating quality and quantity of cDNA, respectively, as we do for all our cDNA work in the program. Edits have been made in the text. 

Reviewer’s comment: Line 223: "or above 2.20".

Response: Edit incorporated. Thank you. 

Reviewer’s comment: Line 229: How many technical replicates were run for qRT-PCR?

Response: Four technical replicates were used for each sample. The sentence has been edited to reflect the number of technical replicates used. 

Reviewer’s comment: Line 272: What condition was defined as the calibrator (to normalize the expression data)?

Response: The parameters inherent in the Normfinder software was used to define the calibrator or data normalization. The Normfinder uses a mathematical model to calculate a stability value from combining the estimates of both the intra- and intergroup variation. Since this is a standard practice, reference to primary literature was incorporated in the text, which is as follows: “Expression of individual genes was normalized in reference to the geometric mean of Pyrus communis �-tubulin and RELATED TO UBIQUITIN1 (RUB1) Cq values, identified as ideal reference genes with NormFinder (Andersen et al., 2004; Imai et al., 2014; Vandesompele et al., 2002) (Supplementary file 6).”

Reviewer’s comment: Comments about the intro: Since the manuscript is titled: "Evidence for pre-climacteric activation of AOX transcripcion during cold-induced...", I would recommend adding additional background about the role of this gene in the introduction. What it is currently present may not enough.

Response: We have added additional text expanding on the role of AOX and alternative respiration and climacteric ripening. 

Reviewer #2: 

Reviewer’s comment: The manuscript entitled “Evidence for pre-climacteric activation of AOX transcription during cold-induced conditioning to ripen in European pear (Pyrus communis L.)” contributes significantly towards understanding of the species- specific fruit ripening process of European pear and has direct implications in preventing post-harvest losses in P. communis. The work is well-performed with defined objectives, appropriate methodology, novel results with statistical significance. The manuscript is well-written with organized figures and tables, and it merits publication with some minor corrections:

Response: Thank you for the encouraging comments. A point-by-point response follows:

Reviewer’s comment: 1. The sample collected for gene expression studies is unclear. Why the expression has to be studied in the peel tissue and not the other part(s)?

Response: Please refer to the response provided above. This sentence has been corrected in the text. Since the tissue was peeled, it contained both peel and flesh tissues. Similar tissue has been used for RNAseq work in pears in previous publications (Nham et al. 2015; Nham et al. 2017).

Reviewer’s comment: 2. In line 172, it is mentioned as ‘The fruit was evaluated for physiological parameters’; however, it is unclear what are the parameters studied? The description is available only for ‘fruit firmness’.

Response: ‘Physiological parameters’ was replaced with ‘firmness’. 

Reviewer’s comment: 3. It is unclear how many replicates were maintained for qPCR analysis. Also, the statistical treatment corresponding to it has to be described.

Response: This has been clarified. Four technical replicates were utilized for qPCR analysis. Details regarding how the statistical treatment was performed using the LinReg software, NMDS analysis and 2-delta-delta Ct using Pfaffl method. Comparative analyses and visualization details are also provided in detail with reference to primary literature. 

Reviewer’s comment: 4. Although the manuscript is well written, the list of references and the in-text citations are not proper in many places. All the references in the reference list should be as per the journal format. In many positions cited references has not been included in the reference list. Those should be incorporated.

Response: Thank you for your effort to point these out. All the edits have been incorporated as advised. 

Reviewer’s comment: 5. Line 109: Place 43 and 44 within the same bracket.

Response: Edit incorporated as advised. 

Reviewer’s comment: 6. In the materials and methods, in many places, reference number has not been mentioned for the in text citations. Line 170: Replace ‘Sugar and Einhorn, 2011’ by reference number 47.

Response: Edit incorporated as advised.

Reviewer’s comment: 7. In Line 190: the reference ‘Gasic et al., 2004’ is missing in the list of references. In text citation should be numbered.

Response: Edit incorporated as advised.

Reviewer’s comment: 8. Line: 192: DNAse will be ‘DNase’

Response: Edit incorporated as advised.

Reviewer’s comment: 9. Line: 222: Both the references: ‘Ramakers et al., 2003’ and ‘Ruijter et al., 2009’ are not present in the list of references. Their corresponding in text citations should be numbered.

Response: Edit incorporated as advised.

Reviewer’s comment: 10. Line: 231: Pfaffl, 2001: Reference should be corrected and corresponding reference number to be mentioned in the text.

Response: Edit incorporated as advised.

Reviewer’s comment: 11. Line 233: ‘Andersen et al., 2004; Imai et al., 2014; Vandesompele et al., 2002’ all of these references are not present in the list of references.

Response: Edit incorporated as advised.

Reviewer’s comment: 12. Line 236: ‘Altschul et al., 1990; Gish and States, 1993’ these references are also missing in the list of references.

Response: Edit incorporated as advised.

Reviewer’s comment: 13. Line 238: mention the corresponding reference number (#77) for ‘Rieu and Powers, (2009)’.

Response: Edit incorporated as advised.

Reviewer’s comment: 14. Line: 272: Use symbols for ‘2-delta-delta Ct’

Response: Edit incorporated as advised.

Reviewer’s comment: 15. Line 301, 303: Mention the reference number (#47) instead of ‘Sugar and Einhorn, 2011’

Response: Edit incorporated as advised.

Reviewer’s comment: 16. Line 317: ‘Wayland, 2003’ reference missing in the list of references.

Response: Edit incorporated as advised.

Reviewer’s comment: 17. Line 319: Replace ‘Kruskal, 1964’ by reference # 64, Replace ‘Krzywinski and Altman, 2014’ by reference # 66. The reference ‘Young, 1970’ is missing in the list of references.

Response: Edit incorporated as advised.

Reviewer’s comment: 18. Line 350: ‘Olsvik, Søfteland’ will be ‘Olsvik et al.’

Response: Edit incorporated as advised.

Reviewer’s comment: 19. Line 423: ‘Nham, Macnish’ will be ‘Nham et al.’

Response: Edit incorporated as advised.

Reviewer’s comment: 20. Line 496: Mention the reference # 87 replacing ‘El-Sharkawy, 2004’

Response: Edit incorporated as advised.

Reviewer’s comment: 21. Line 509: All of the references ‘Chang et al., 2014; Ma et al. 2012; Qiu et al., 2012’ are missing in the list of references.

Response: Edit incorporated as advised.

Reviewer’s comment: 22. Line 537: Mention the reference # 87 replacing ‘El-Sharkawy, 2004’

Response: Edit incorporated as advised.

Reviewer’s comment: 23. Line 547: Replace ‘Sivankalyani et al., 2014’ by reference # 90.

Response: Edit incorporated as advised.

Reviewer’s comment: 24. Line: 565: Replace ‘Liu et al., 2013’ by reference # 12

Response: Edit incorporated as advised.

Reviewer’s comment: 25. Line 566: Replace ‘Tacken et al. 2012a’ by reference # 58. 

References ‘Robles et al., 2012; Schaffer et al., 2013;’ are not present in the list of references.

Response: Edit incorporated as advised.

Reviewer’s comment: 26. Line 618: Mention reference # 68 within ()

Response: Edit incorporated as advised.

Reviewer’s comment: 27. For the list of references, the formatting should be corrected wherever they are not as per the journal format

Response: Edit incorporated as advised.

Reviewer’s comment: 28. Line: 727: Rewrite the reference as per journal format

Response: Edit incorporated as advised.

Reviewer’s comment: 29. Line 835: Rewrite the reference as per journal format

Response: Edit incorporated as advised.

Reviewer’s comment: 30. Line 880: Rewrite the reference as per journal format

Response: Edit incorporated as advised.

Reviewer’s comment: 31. Line 916: Rewrite the reference as per journal format

Response: Edit incorporated as advised.

Reviewer’s comment: 32. Line 925: Correction needed

Response: Edit incorporated as advised.

Reviewer’s comment: 33. Line 929: Rewrite the reference as per journal format

Response: Edit incorporated as advised.

Reviewer’s comment: 34. Line 966: Rewrite the reference

Response: Edit incorporated as advised.

Reviewer’s comment: 35. Line 1012: Rewrite the reference

Response: Edit incorporated as advised.

Reviewer’s comment: 36. A minor polishing in terms of grammar is required.

Response: The authors read through the manuscript and a few grammatical edits were incorporated. 

Nham NT, de Freitas ST, Macnish AJ, Carr KM, Kietikul T, Guilatco AJ, Jiang C-Z, Zakharov F, Mitcham EJ (2015) A transcriptome approach towards understanding the development of ripening capacity in 'Bartlett' pears (Pyrus communis L.). Bmc Genomics 16:762-762. doi:10.1186/s12864-015-1939-9

Nham NT, Macnish AJ, Zakharov F, Mitcham EJ (2017) ‘Bartlett’ pear fruit (Pyrus communis L.) ripening regulation by low temperatures involves genes associated with jasmonic acid, cold response, and transcription factors. Plant Science 260:8-18. doi:https://doi.org/10.1016/j.plantsci.2017.03.008

---

## [Decision Letter · Decision Letter 1]

15 Nov 2019

Evidence for pre-climacteric activation of AOX transcription during cold-induced conditioning to ripen in European pear (Pyrus communis L.)

PONE-D-19-25331R1

Dear Dr. Dhingra,

We are pleased to inform you that your manuscript has been judged scientifically suitable for publication and will be formally accepted for publication once it complies with all outstanding technical requirements.

With kind regards,

Muthamilarasan Mehanathan, Ph.D.

Academic Editor

PLOS ONE

Additional Editor Comments (optional):

Reviewers' comments:

Reviewer's Responses to Questions

**Comments to the Author**

1. If the authors have adequately addressed your comments raised in a previous round of review and you feel that this manuscript is now acceptable for publication, you may indicate that here to bypass the “Comments to the Author” section, enter your conflict of interest statement in the “Confidential to Editor” section, and submit your "Accept" recommendation.

Reviewer #1: All comments have been addressed

Reviewer #2: All comments have been addressed

2. Is the manuscript technically sound, and do the data support the conclusions?

Reviewer #1: Yes

Reviewer #2: Yes

3. Has the statistical analysis been performed appropriately and rigorously? 

Reviewer #1: Yes

Reviewer #2: Yes

4. Have the authors made all data underlying the findings in their manuscript fully available?

Reviewer #1: Yes

Reviewer #2: Yes

5. Is the manuscript presented in an intelligible fashion and written in standard English?

Reviewer #1: Yes

Reviewer #2: Yes

6. Review Comments to the Author

Reviewer #1: The manuscript presents interesting data regarding ripening in pear. The study is technically and scientifically sound. A few aspect required some clarification, however, these were properly addressed by the authors.

Reviewer #2: (No Response)

7. PLOS authors have the option to publish the peer review history of their article (what does this mean?). If published, this will include your full peer review and any attached files.

Reviewer #1: No

Reviewer #2: No

---

## [Editor Report · Acceptance letter]

22 Nov 2019

PONE-D-19-25331R1 

Evidence for pre-climacteric activation of AOX transcription during cold-induced conditioning to ripen in European pear (Pyrus communis L.) 

Dear Dr. Dhingra:

I am pleased to inform you that your manuscript has been deemed suitable for publication in PLOS ONE. Congratulations! Your manuscript is now with our production department. 

With kind regards,

on behalf of

Dr. Muthamilarasan Mehanathan 

Academic Editor

PLOS ONE